# Interleukin 10 (IL-10) Production and Seroprevalence of *Entamoeba histolytica* Infection among HIV-Infected Patients in South Africa

**DOI:** 10.3390/pathogens12010019

**Published:** 2022-12-23

**Authors:** Renay Ngobeni, Jeffrey Naledzani Ramalivhana, Afsatou Ndama Traore, Amidou Samie

**Affiliations:** 1Department of Biochemistry and Microbiology, University of Venda, Thohoyandou 0950, Limpopo, South Africa; 2Department of Medicine, Division of Infectious Diseases, University of North Carolina School of Medicine, Chapel Hill, NC 27599, USA; 3SAMRC-UNIVEN: Antimicrobial and Global Health Research Unit, University of Venda, Thohoyandou 0950, Limpopo, South Africa

**Keywords:** *Entamoeba histolytica*, intestinal parasites, genetic susceptibility, HIV/AIDS, seroprevalence, interleukin 10, antigen detection

## Abstract

Infections by the parasite *E. histolytica* are increasing in HIV-infected individuals. Interleukin (IL-10) plays an important role in maintaining the mucosal barrier. Therefore, the seroprevalence of *E. histolytica* was investigated in relation to the IL-10 serum concentration among HIV- infected patients. A total of 647 blood samples were collected from asymptomatic HIV-infected patients. The *Entamoeba histolytica* antigen (GALNAC lectin) and serum antibodies were assessed using specific ELISAs (TECHLAB, Virginia, USA). IL10 blood levels were measured using a commercial ELISA test, and the results were analyzed using parametric and non-parametric statistical tests. The Gal/GALNAC lectin was detected in only 0.5% (3/647) of individuals, and the antibodies against *E. histolytica* were detected in 65.2% (422/647) of the samples. A significant increase in IL-10 levels was found in 68.1% of patients who were sero-negative for *E. histolytica* antibodies compared to patients who were sero-positive. There is a high level of exposure to *E. histolytica* among HIV patients in South Africa, although the prevalence of amoebic liver abscesses might be low. This study revealed that elevated levels of IL-10 might be associated with a reduced risk of amebiasis.

## 1. Introduction

Diarrheal diseases are a significant problem worldwide, especially in low- and middle-income countries [1]. Amebiasis, caused by infection with *Entamoeba histolytica (E. histolytica)*, is documented as one of the most problematic parasitic diseases in both developing and developed countries, especially in places of poor sanitation and inadequate water supply [2]. In addition, amebiasis is the third leading cause of death among parasitic diseases, after malaria and schistosomiasis [3]. It has been reported to infect about 10% of the world’s population, resulting in 50 million cases of invasive amebiasis (colitis and liver abscesses). Although most subjects infected with *E. histolytica* are asymptomatic, it accounts for more than 100,000 deaths annually [4]. Moreover, although some individuals are asymptomatic when infected with amoeba, they can transmit the parasites to susceptible persons. 

Amebiasis is increasing in HIV-infected individuals [5]. It is suggested that HIV-infected patients are more susceptible to an invasive form of the disease by *E. histolytica* than HIV-negative patients. However, it is still unclear whether HIV is a risk factor for *E. histolytica* infections. Opportunistic infections, such as parasites, are known to intensify the condition of HIV-infected patients, and they play a significant role as the most common cause of morbidity and mortality in HIV patients [5]. Diarrhea is one of the most common presenting complaints in HIV-infected patients, where most diarrheal infections are associated with intestinal parasites [6]. Numerous studies demonstrated that intestinal parasites, such as *Cryptosporidium* spp., *Microsporidia* spp., *Isospora belli*, and *Cyclospora cayetenensis*, were frequently associated with episodes of severe and often fatal diarrhea in HIV-infected patients [7,8,9]. Among these parasites, *Entamoeba histolytica*, *Giardia intestinalis*, and *Strongyloides stercoralis* are documented as the most important cause of diarrhea in immune-compromised patients. Various countries have reported a high prevalence of amebiasis in HIV/AIDS patients. A significant association between high levels of the serum anti-*E. histolytica* antibodies and the presence of *E. histolytica* in the stool have been noted in various studies. A study in South Africa by Samie [10], in the Vhembe District of South Africa, has indicated a positive association between *E. histolytica* infection and HIV-positive individuals. 

It has been shown that HIV can alter cytokine production and responsiveness in order to increase virus production and hinder the immune response [5]_._ Cytokine dysregulation is central to HIV infection; it can regulate the replication of HIV and also the development of immunodeficiency [11]. Interleukin (IL-10) is an important anti-inflammatory cytokine, which plays an important role in this cytokine dysregulation since it is able to down-regulate cytokine synthesis in monocytes, macrophages, and Th1 lymphocytes [12]. A possible overproduction of IL-10 in HIV patients with a down-regulation of Th1 responses and an inhibition of various macrophage functions might contribute to the increased susceptibility to infections with various parasites, bacteria, and viruses. 

In most cases, *E. histolytica*-infected individuals do not develop symptoms. The most common clinical symptom and signs of amoebic liver abscesses are abdominal pain, fever and chills, and abdominal tenderness. However, in most cases, there are no significant differences in the presentation of the disease between HIV-positive and -negative patients [5]. Several studies have led to an improved understanding of the host immune response in intestinal amebiasis. However, the question remains unanswered; for example, the host-dependent immune mechanisms that might be responsible for the development of severe diseases are not fully understood. Many individuals have a different genetic signature that could affect how quickly and effectively the immune system can act to buildup resistance to infections. Interleukin-10 is an important immunoregulatory cytokine [11] that serves as a pre-determining factor for amebic invasion [12]. A study by Hamano [12] has shown that IL-10 is required for resistance to intestinal amebiasis and that IL-10 deficiency leads to high susceptibility to amebic infection in mice. However, the impact of IL-10 production on amebiasis–HIV coinfection is not known. Therefore, the present study sought to elucidate the role of IL-10 during amebic infection in HIV-positive and -negative patients.

## 2. Materials and Methods

### 2.1. Study Site, Data, and Sample Collection

Blood samples were collected from asymptomatic HIV-positive patients in Polokwane (Limpopo Province, South Africa). A questionnaire was used to collect demographic and clinical information, such as origin, age, sex, the cluster of differentiation 4 (CD4) count, and other medical conditions, such as sexually transmitted diseases. Trained personnel collected six hundred and forty-seven (647) blood samples from HIV/AIDS patients who were attending different hospitals and clinics in 3 districts (Capricorn, Sekhukhune, and Waterberg) in the Limpopo Province. The blood was collected in EDTA tubes and transported to the Parasitology laboratory of the University of Venda’s Department of Microbiology for further analysis. Upon arrival at the laboratory, the blood was centrifuged for the collection of the plasma, buffy coat, and red blood cells. The samples were aliquoted and stored at −20 °C until further analysis. 

### 2.2. Detection of E. histolytica Antibodies in Serum Specimens by ELISA 

The *E. histolytica* serology’s ELISA was supplied by TechLab, Inc. (Blacksburg, VA, USA) and was specifically designed to identify antibodies against *E. histolytica* in serum samples. For this study, the serum samples were diluted by 1:50 using the diluent supplied with the kit. Briefly, 100 µL of the diluted serum was added to each sample well, and a drop (50 µL) of the positive control and 100 µL of the diluent (negative control) were added to the control wells. The plate was then covered with an adhesive sheet and incubated for an hour at room temperature. After incubation, the plate was washed four times with 1× a wash solution. One drop of the conjugate was added into each well, sealed with an adhesive sheet, and incubated for 30 min at room temperature. After incubation, the plate was washed, as described above. Two drops of the substrate were added to each well and incubated at room temperature for ten minutes. After the incubation step, one drop of a stop solution was added to each well, and the stop solution converted the blue color to yellow. The absorbance was measured at 450 nm on a VersaMax™ microplate ELISA reader (Molecular Devices, Sunnyvale, CA, USA). Samples with an OD greater than 0.150 were considered positive. 

### 2.3. Detection of E. histolytica Antigens in Serum Specimens by ELISA

A Techlab *E. histolytica* test II test (Techlab, Inc, Blacksburg, VA, USA) was used for the detection of the *E. histolytica* antigen (Gal/galnac lectin). According to the manufacturer’s instructions, the test is intended for use on fecal specimens. However, in the present study, the kit was used for the detection of adhesion in serum samples; therefore, some of the steps were optimized. Briefly, 100 µL of the serum specimen was diluted with 100 µL of the sample diluent. One drop of the conjugate was added to each well, and the samples and controls were then added to wells containing the conjugate; the plate was covered with an adhesive sheet and incubated for 2 h at room temperature. After the washing step, two drops of a substrate reagent were added to all the test wells and incubated at room temperature for 10 min. After the incubation period, one drop of the stop solution was added; the stop solution changed the blue color to yellow, and after two minutes, the optical density was read at 450 nm on the VersaMax™ microplate ELISA reader (Molecular Devices, Sunnyvale, CA, USA). The specimen was considered positive if the reading was greater than 0.050. 

### 2.4. Detection of IL-10 Concentration in Serum Specimens by ELISA

The concentration of IL-10 was measured in the serum samples using the ELISA method from MABTECH (Nacka Strand, Sweden). On day 1, the plate was coated by adding 100 µL of the capture antibody to each well (monoclonal antibody 9D was diluted to 2 µg/mL in a PBS buffer) and incubated at 4–8 °C overnight. At day 2, the plate was washed two times with the PBS and blocked with a blocking buffer containing 0.05% of Tween 20 and 0.1% BSA; the plate was incubated for an hour at room temperature. After the incubation period, the plate was washed five times with a buffer containing 0.05% Tween. The samples and standards were diluted with an incubation buffer, and 100 µL of the samples and the standards were added to each well and incubated for 2 h at room temperature. After the wash step, 100 µL microliters of the detection antibody (biotinylated monoclonal antibody 12G8) was added into each well and incubated for 1 h at room temperature. Following the wash step, 100 µL of the streptavidin–HRP was added into each well and incubated for an hour at room temperature. After washing the plate, two drops of the substrate were added into each plate well, and the plate was incubated for 30 min at room temperature. One drop of the stop solution was added to each well and read within 10 min at 450 nm on the VersaMax™ microplate ELISA reader (Molecular Devices, Sunnyvale, CA, USA).

### 2.5. Statistical Analysis

The results were entered into an excel spread sheet and edited appropriately (Microsoft office package) and analyzed using the Statistical Package for the Social Sciences (SPSS for WINDOWS version 18.0). Assuming that the data followed a normal distribution, the comparison of proportions and statistical significance were tested using a Chi-square test. A *p*-value of <0.05 was considered statistically significant

## 3. Results

### 3.1. Demographic Characteristics of the Study Population

In total, 647 patients were included in this study. The patients’ ages ranged from 1–68 years; their mean age was 35.86 ± 12.75. Most patients lived in rural areas (427; 66%), where the hygiene standards may have been low. One hundred and ninety-seven were males, and four hundred and forty-six were females. Out of these participants, 332 (51.3%) were from the Capricorn District, 169 (26%) were from the Sekhukhune District, and 132 (20.4%) were from the Waterberg District. Most patients (36.3%) had a CD4 count ranging between 251 and 500 cells/mm³ (Table 1). 

### 3.2. Distribution of E. histolytica Infection in the Study Population

The results of this study showed that a sero-prevalence of *E. histolytica* was high in the study population, as 422 (65.2%) of the tested subjects had antibodies against *E. histolytica*. The present study also showed a low prevalence of the E. histolytica Gal/GalNAc lectin in the study population since only three (0.5%) out of six hundred and forty-seven patients were found to have antigens against E. histolytica. 

The samples were collected from three different districts in Limpopo Province. The antibodies against *E. histolytica* were found to be the highest in Waterberg (67.4%), compared to Capricorn (65.4%) and Sekhukhune (64.5%). The prevalence was high in rural areas (66.5%) compared to urban areas (63.9%). The people who were attending hospitals were more infected (73.8%) than the people in clinics (61.5%), and the difference was significant (*p* < 0.005). Females were more infected (66.1%) than males (63.5%). Patients aged between 0 and 25 had a high prevalence of 68.9% compared to the other age groups. The sero-prevalence of E. histolytica was much higher in patients with a CD4 count higher than 1000 (69.7%) than in patients with a CD4 count lower than 1000. All the results are shown in Table 2 below.

### 3.3. The Prevalence of E. histolytica Antigens in the Study Population 

The Techlab *E. histolytica* II test detected antigens against *E. histolytica* in three (0.5%) out of the six hundred and forty-seven serum samples analyzed in this study. All three patients were females, two from Capricorn and one from Sekhukhune. All of them were from rural areas. These patients were tested for the IL-10 concentration in the serum, and they all showed low concentrations of IL-10 in their blood. Two of these patients were sero-positive for the antibodies against *E. histolytica* (Table 3). 

### 3.4. Distribution of Interleukin-10 Levels in the Serum Samples

A total of 647 serum samples were tested for the concentration of IL-10 in the serum samples by the ELISA tests from MABTECH. The serum samples from both *E. histolytica* sero-positive and sero-negative patients were used in this study. The sero-negative patients were used as a control group. The expression of IL-10 was compared in relation to the age of the study participants, and it was found that patients aged between 46 and 80 (35.3%) produced high IL-10 levels, followed by age groups 0 to 25 (27.9%), and the least expression of this cytokine was obtained in people aged 26 to 45, with a percentage of 27.3%. Females were found to produce higher levels of IL-10 (30.3%) compared to males (25.9%) (Table 4).

### 3.5. The Distribution of IL-10 Levels among E. histolytica Sero-Positive and Sero-Negative Patients in the Study

The percentage of *E. histolytica* infection was higher among patients who had a low concentration of IL-10 (68.1%), while in patients with high levels of IL-10, the percentage of the infection was lower (58.2%). Therefore, IL-10 was protective against *E. histolytica* infections (Figure 1). The patients with a CD4 count greater than 1000 had a high percentage of IL-10 expression than those with a CD4 count between 0 and 250 (29.2%), 251 and 500 (23.4%), and 500 and 1000 (34.3%). The study showed that the higher the level of the CD4 count, the higher the prevalence of *E. histolytica* infection. However, the difference was not statistically significant (Figure 2). 

## 4. Discussion

Amebiasis is an intestinal and extra-intestinal disease caused by the protozoan parasite *E. histolytica*, which, amongst all the *Entamoeba* species, is the only species that is considered to be pathogenic to humans. The infection has been reported to occur worldwide but is prevalent in tropical and developing countries. The disease affects major organs, such as the colon and liver, with different symptoms or is sometimes asymptomatic [13]. However, the question of what causes the difference between disease presentation and asymptomatic infection is still unanswered. It has been suggested that both the parasite and host genetics play a role in the outcome of amebic infection. 

In the present study, two ELISA techniques were used. These included antigen-detecting and antibody detection. The antigen detection ELISA kits have been used for the diagnosis of amebic liver abscesses in previous studies [14]. The detection of antibodies often indicates a previous encounter of the individual with the parasite and might not necessarily indicate symptomatic disease [15]. The findings of the present study indicate that amebiasis is highly prevalent in the study population. Of the 647 samples tested, 422 (65.2%) were sero-positive for anti-*E. histolytica* antibodies. The sero-prevalence of amebiasis in this study was higher compared to the study conducted in 2010 by Samie [10], as well as the one conducted in Pakistan, where the sero-prevalence of amebiasis was found to be 73% [16]. A study in the Nasarawa State in Nigeria reported a 40.0% prevalence of intestinal amebiasis in school-age children in Lafia [17]. A study by Zhou [18] showed a prevalence of 41.1% of *E. histolytica* infection among the MSM community from China, showing that the infection is also transmitted through sexual contact in men who have sex with men. 

The present study showed that females (66.1%) were more affected than males (63.5%), although the difference was not statistically significant. Some observations have explained the reasons for this difference in the prevalence between males and females in that most families favor female housemaids over males for cultural, religious, and traditional reasons [19]. Similar results on the distribution of *E*. *histolytica* by gender were observed in a study by Samie [10]; this study showed that *E. histolytica* infection was higher in females than in males, demonstrating that females are more likely than males to carry asymptomatic infections rather than developing invasive diseases. The findings of this study are also in support of a study by Jamila [20], who found similar results of higher *E. histolytica* infection in females (48.7%) than in males (47.8%). 

Although all age groups were sero-positive for *E. histolytica* antibodies in the present study, the patients aged between 0 and 25 showed a very high prevalence (68.9%), followed by the older age at 46–80 years, with a prevalence percentage of 67.6%, and the least percentage was observed in patients aged between 26 and 45 (64.0%). Several studies have found similar results. A study in Jeddah on the factors associated with a high prevalence of *Entamoeba histolytica*/*dispar* infection among children showed a high prevalence of *E. histolytica* infection among children aged 1 month- 6 years, with a prevalence of 60.8%. However, their study only focused on children with a sample size of 300 [20]. The results of the present study also showed a high prevalence of *E. histolytica* in participants aged 46–80, which is in agreement with previous results reported by Samie [10] in the Limpopo Province. Tasawar and colleagues [21] reported that *E. histolytica* infection was highest in the age group of 33 to 48 years (16.67%).

The sero-prevalence of *E. histolytica* is said to be high in areas with poor sanitation, inadequate water supply, and low socioeconomic development, especially in developing countries. These observations are supported by the results of the present study. Our study found that the patients in rural areas were more sero-positive (66.0%) than the patients in urban areas (63.9%). The high rates in rural areas may be influenced by improper hygiene and agricultural backgrounds [22]. Furthermore, we found that a higher sero-prevalence was obtained for patients attending hospitals compared to those attending clinics. Concerning the correlation between the *E. histolytica* infection and CD4 count, the results of this study showed no significant association between *E. histolytica* and the CD4 count. The three individuals reported to have ‘other medical conditions’, such as kidney and liver damage, were found sero-positive for *E. histolytica*, showing that the patients in this study had mixed infections. There is a possibility that the patients presenting liver damage can encounter amebic liver abscesses in the future.

An amebic liver abscess is the most common extra-intestinal manifestation of amebiasis. It has been described as a tropical disease or a disease that is less prevalent in temperate climates, where it is also known as a summertime disease [23,24]. However, its epidemiology and socio-demographic determinants are not well described in the literature. In this study, there were only three (0.5%) patients who showed the *Entamoeba* antigen in their blood, indicative of extra-intestinal amebic infection, such as amebic liver abscesses. The observations of the present study are in agreement with the results obtained in a study by MecGarr [24], where only two patients were found with proven amoebic colitis. The low sero-positivity rate of the Entamoeba antigen (0.5%) recorded in the present study is an indication that this extra-intestinal amebiasis might be uncommon in our study population. 

It has been documented that an amebic liver abscess evolves to fulminant necrotizing rupture or colitis in approximately 0.5% of cases [25], and the mortality rate due to amebic liver abscesses has fallen to 1–3% in the last century, following appropriate or effective treatment. A study by Zeehaida [26] explained that the diagnosis of ALA is sometimes difficult since its clinical manifestations are highly different. Although the Techlab *E. histolytica* II kit is meant for the detection of antigens against *E. histolytica* in stool samples, it has been used for the detection of the amebic antigen in the blood, and our study has shown that this kit could also be used with serum samples because of its usefulness for the detection of ALA. However, some studies have found the kit not useful for ALA detection [26].

Cytokines are important mediators of the host’s immune response. They can be used as markers to determine the immune mechanisms underlying the outcome of amebic infection [27]. Despite the high prevalence of intestinal amebiasis, the host’s innate immune response, as it relates to the clinical outcomes of amebic infection, remains poorly characterized in humans, especially in the Limpopo Province. Therefore, this study attempted, for the first time, to integrally evaluate the potential effect of IL-10 expression on the disease outcome in Limpopo Province.

Out of the 467 serum samples tested for IL-10 levels in the present study, 225 were negative for *E. histolytica* antibodies, while 422 were positive for *E. histolytica* antibodies. Very few studies have associated IL-10 levels with amebiasis. A study by Bernin [27] showed that IL-10 serum levels were significantly increased in ALA patients, indicating that the invasion of the liver tissue by *E. histolytica* elicits an anti-inflammatory immune response. In the present study, mostly asymptomatic patients were used. Even though many of the participants were positive for *E. histolytica* antibodies, most of these had lower IL10 levels, while those that were sero-negative had higher IL10 levels. This is potentially an indication that the IL10 level is protective of *E. histolytica*. The current study also showed that patients with high levels of IL-10 are likely to have a low burden of amebiasis compared to patients who produce low levels of IL-10. The present study also confirms a previous study by Hamano and colleagues who showed that IL10 protected C57BL/6 mice from amebiasis [12]. The elevated levels of IL-10 show that the patient is already having inflammation and that the body is trying to bring the inflammation down. This indicates the possibility of a protective role of IL-10 in patients who might have been infected with *E. histolytica*. Therefore, elevated levels of IL-10 in the serum may play a significant role in the development of amebiasis. 

## 5. Conclusions

The present study demonstrated a high sero-prevalence of *E. histolytica* with an apparently low prevalence of extra-intestinal amebiasis. About 65% of individuals living in Polokwane in Limpopo Province have been in contact with *E. histolytica*. These findings show that intestinal amebiasis is very common in the study population, while amebic liver abscesses might be uncommon (0.5%). Elevated levels of IL-10 in patients may serve as protection against the parasite. The higher the level of IL-10 in the serum, the lower the infection (sero-positivity). 

The occurrence of *E. histolytica* in this study is of public health significance, and if appropriate care is not taken, it could result in epidemic situations. Amebiasis can affect anyone; however, the disease mostly occurs in young and middle-aged adults. The observation that this parasite affected the young and old people in this study opens a new dimension as to the source of infection in old age and young groups in the African setting. This study also observed that the infection is more prevalent in rural areas than in urban. It is, therefore, recommended that the old-aged and the young group, as well as people in rural areas, be educated properly on hygiene and especially more on personal hygiene.

## Figures and Tables

**Figure 1 pathogens-12-00019-f001:**
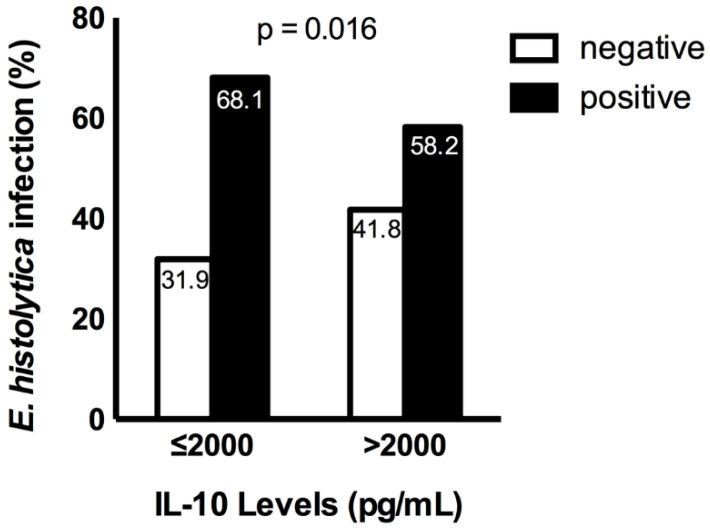
The distribution of *E. histolytica* among patients with lower IL-10 concentrations compared to those with higher concentrations of IL-10.

**Figure 2 pathogens-12-00019-f002:**
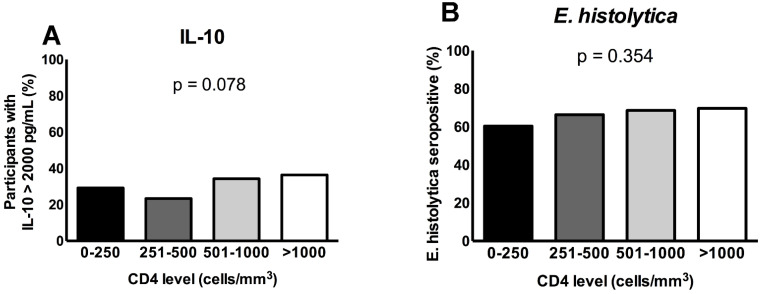
(**A**) The production of IL-10 in relation to the CD4 count of the patients. (**B**) *E. histolytica* sero-positivity in relation to CD4 counts’ range.

**Table 1 pathogens-12-00019-t001:** Demographic characteristics of the patients in the study.

Characteristics		Frequency	Percent (%)
**Origin**	Capricorn	332	51.3
	Sekhukhune	169	26.1
	Waterberg	132	20.4
	Missing system	14	2.2
**Area**	Urban	216	33.4
	Rural	427	66.0
	Missing system	4	0.6
**Sector**	Hospital	195	30.1
	Clinic	434	67.1
	Missing	18	2.8
**Gender**	Male	197	30.4
	Female	446	68.9
	Missing system	4	0.6
**Age group**	0 to 25 years	61	9.4
	26 to 45 years	439	67.9
	46 to 80 years	136	21.0
	Missing system	11	1.7
**CD4 count**	0 to 250	192	29.7
	251 to 500	235	36.3
	501 to 1000	172	26.6
	More than 1000	33	5.1
	Missing	15	2.3
**Total**		647	100

**Table 2 pathogens-12-00019-t002:** Distribution of *E. histolytica* antibodies in the study population.

Characteristics		*E. histolytica* Antibodies (Seropositive)	Statistics
**Origin**	Capricorn	217 (65.4%)	*p* = 0.863
	Sekhukhune	109 (64.5%)	
	Waterberg	89 (67.4%)	
**Area**	Rural	138 (63.9%)	*p* = 0.588
	Urban	282 (66.0%)	
**Sector**	Hospital	144 (73.8%)	*p* =0.003
	Clinic	267 (61.5%)	
**Gender**	Male	125 (63.5%)	*p* = 0.509
	Female	295 (66.1%)	
**Age groups**	0–25	42 (68.9%)	*p* = 0.609
	26–45	281 (64.0%)	
	46–80	92 (67.6%)	

**Table 3 pathogens-12-00019-t003:** Characteristics of three patients positive for antigens against *E. histolytica*.

Sample Code	Age	Gender	Origin	Area	CD4 µL/mL	*E. histolytica* Antibodies	*E. histolytica* Antigens	Il-10 Level
POI0840	64	Female	Sekhukhune	Rural	338	Positive	Positive	<2000 pg
POI0846	29	Female	Capricorn	Rural	398	Positive	Positive	<2000 pg
POI1019	49	Female	Capricorn	Rural	373	Negative	Positive	<2000 pg

**Table 4 pathogens-12-00019-t004:** The distribution of IL-10 levels in relation to age and gender of the study participants.

Characteristics		IL-10 Expression		Statistics
		<2000 pg/mL	>2000 pg/mL	
**Age group**	0 to 25	44 (72.1%)	17 (27.9%)	χ^2^ = 3.238, *p* = 0.198
	26 to 45	319 (72.7%)	120 (27.3%)	
	46 to 80	88 (64.7%)	48 (35.3%)	
**Gender**	Male	146 (74.1%)	51 (25.9%)	χ^2^ = 6.980, *p* = 0.222
	Female	311 (69.7%)	135 (30.3)	

## Data Availability

All data presented in this article are available upon request to corresponding author (Samie A.).

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
