# Peer review of "Interleukin 10 (IL-10) Production and Seroprevalence of Entamoeba histolytica Infection among HIV-Infected Patients in South Africa"

_pathogens, 2022, doi:10.3390/pathogens12010019_

Round 1

Reviewer 1 Report

The manuscript is an important investigation in the immunological variables of HIV patients regarding the risk of the infection due to E. histolytica parasite that was not previously investigated 

Author Response

We are grateful for the positive comments.

Author Response

We addressed the comments on the attached document. We have also considered the recommended reference and added two more references for clarity. 

Reviewer 3 Report

This manuscript is well-written and the results are clearly and appropriately presented. 

There are no suggestions for changes.

Author Response

(The authors gave the same response as above.)

Round 2

Author Response

To the Editor: Pathogens

Re:  Manuscript ID: pathogens-2019744

Title: Interleukin 10 (IL-10) Production, and Seroprevalence of Entamoeba histolytica infection among HIV-infected patients in South Africa.

Dear Editor,

In response to your and the reviewers’ comments concerning our manuscript submitted to Pathogens, we have addressed the suggestions as follows: The responses are indicated in red to differentiate them from the reviewers’ comments:

Re-Reviewer 2:

Ngobeni R, et al. modified their manuscript according to the reviewer’s recommendation. The authors should address some points and make modifications about the interpretations for IL-10 levels among E. histolytica sero-positive and -negative participants in this study.

Major comments: 1. In results 3.4. what is the meaning of “the overall IL-10 expression”?

The phrase has been deleted from the text as it creates confusion and doesn’t seem to be emanating directly from the results presented in the table (Table 4).

In my understanding, the authors checked IL-10 level from 647 participants, which consisting of 422 E. histolytica seropositives (65.2%) and 225 Ehis seronegatives (34.8%). However, in discussion section (line 385-), they described “E. histolytica sero-positive patients produced more IL-10 than sero-negative patients”. I could not find any supporting data in their results.

The phrases have been removed from the results section and from the discussion to avoid confusion.

On the contrary, Figure 1 showed that seropositivity among high IL-10 group is significantly lower (58.2%) than that among low IL-10 group. This data strongly indicates that IL-10 levels are lower among sero-positive participants (it can be calculated from table 4 & figure 1 that IL-10 elevation is seen in about 65% among seronegatives, and about 25% among seropositives.).

This indeed is our main conclusion.  Our hypothesis is that if samples with high IL10 levels are less positive for E. histolytica, and samples with low IL10 levels are more positive for E. histolytica, then IL10 has help reduced the level of positivity of E. histolytica which basically means that IL10 is protective of E. histolytica.    

Their discussions and conclusions are not reliable at all.

We have now corrected the discussion and conclusions by removing the controversial interpretation.

The authors must clarify whether IL-10 levels are high or low among sero-positive (or negative) participants with the supporting data.

This has now been clarified: As indicated in figure 1, IL10 levels were low in sero-positive samples while they were high in sero-negative samples.

  1. In the abstract, where is the come from the description “a significant increase in IL-10 levels was found in 68.1 % of patients who were positive for E. histolytica antibodies compared to patients who were seronegative.”??? 68.1% can be found in figure 1, however, it presents 68.1% of the participants without IL-10 elevation were sero-positive for E. histolytica. It is inversed results to the description in the abstract. Thus, the author should present IL-10 data correctly, and resubmit the paper before more detail review.

This has now been corrected throughout the text from the abstract to the discussion and conclusions.

Minor comments: 1. Do the authors use plasma instead of serum? In Materials and Methods (2.1), the blood was collected in EDTA tubes (line 89). Please clarify the type of the blood samples. If they used plasma instead of serum for antibody testing (and IL-10), they used correct words throughout of the manuscript, and add some comments about limitations in discussion section.

The samples collected were in EDTA tubes. Therefore, plasma samples are the ones that were used.

We have also added some limitations of the present study in the discussion section.

We Appreciate the comments and criticisms of the reviewers that have help improved our manuscript.

 Sincerely.

The authors.

Round 3

Reviewer 2 Report

No more comments on the revised manuscript from the reviewer.